# TFGNet: Target Face Generation from Low-Quality Images via Textual Guidance

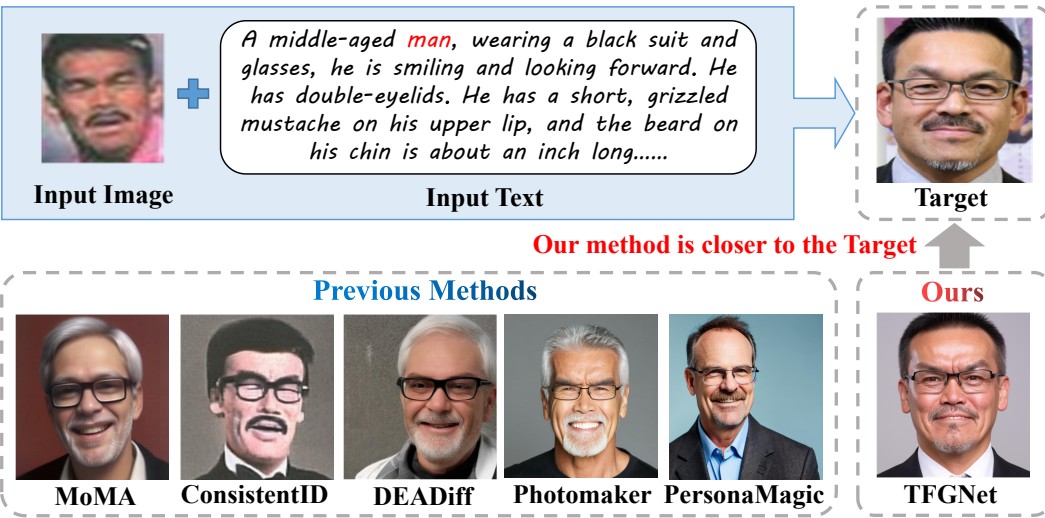

Figure 1: Given a textual description and a low-quality face image as input, the goal is to generate a high-quality image that closely matches the target image. We compare the proposed TFGNet with five state-of-the-art methods: MoMA Song et al. (2024), ConsistentID Huang et al. (2024), DEADiff Qi et al. (2024), Photomaker Li et al. (2024) and PersonaMagic Li et al. (2025). TFGNet achieves superior identity preservation and more faithful detail reconstruction.

## Abstract

Recent advances indicate that text-guided face generation has attracted considerable attention in the field of computer vision. However, most existing methods assume high-quality (HQ) face inputs, while how to generate HQ and identity-preserving faces from low-quality (LQ) images is still an open problem. In this paper, we propose a novel face generation approach named TFGNet, which generates HQ face images from LQ inputs guided by textual descriptions. Unlike most existing methods that depend on HQ inputs, TFGNet leverages external textual descriptions as semantic guidance to directly generate HQ and identity-preserving faces from degraded images. First, we design a unified framework that integrates a Transformer-based encoder, a codebook mechanism, and multimodal representations extracted from contrastive language-image pretraining (CLIP) model to produce enhanced cross-modal embeddings, which are then decoded by a diffusion model to generate target face images with both high visual fidelity and accurate identity (ID) retention. Second, we propose a masked diffusion loss that emphasizes identity-related regions and incorporate it into a dynamically weighted total loss, enabling a balanced trade-off among visual fidelity, semantic coherence, and ID consistency. Third, we build a multimodal dataset comprising LQ face images, HQ targets, and manually annotated textual descriptions to address the scarcity of suitable text-image pairs for this task. Extensive experimental results demonstrate that the proposed TFGNet approach outperforms many state-of-the-art techniques in face generation in terms of both objective metrics and perceptual quality.

## 1 INTRODUCTION

Face images play a vital role in public security applications, particularly in urban surveillance and criminal investigations, as they often provide crucial evidence for case resolution. However, various factors such as complex lighting conditions, adverse weather, limitations of imaging devices, and occlusions frequently lead to low-quality (LQ) face images Chen et al. (2018); Bulat & Tzimiropoulos (2018); LI et al. (2019), which significantly hinder their utility. In response, many studies Nasir et al. (2019); Xia et al. (2021); Wang et al. (2021); Ding et al. (2021); Patashnik et al. (2021); Gal et al. (2022); Li et al. (2022); Sun et al. (2022); Podell et al. (2023); Li et al. (2024); Kong et al. (2024); Ma et al. (2024); Li et al. (2025) have explored cross-modal generation (e.g., text-to-face generation) to recover or reconstruct high-quality (HQ) face images. These efforts have been supported by the rapid development of deep learning-based generative modeling techniques, such as generative adversarial networks (GANs) Nasir et al. (2019); Xia et al. (2021); Patashnik et al. (2021); Gal et al. (2022), Transformers Ding et al. (2021); Wang et al. (2021); Sun et al. (2022); Li et al. (2022), and diffusion models Podell et al. (2023); Li et al. (2024); Kong et al. (2024); Ma et al. (2024); Li et al. (2025), which have improved the quality and controllability of face generation. For example, Sun et al. (2022) proposed a Transformer-based method for generating face images from textual descriptions. Li et al. (2024) introduced a diffusion-based framework that employs stacked identity (ID) embeddings to guide text-driven face generation. Li et al. (2025) presented a diffusion-driven model that leverages stage regulation and a Tandem Equilibrium mechanism to achieve high-fidelity face generation. Despite their impressive performance, most existing methods generate face images from HQ inputs. When confronted with LQ or degraded images, how to generate HQ and identity-preserving faces remains an open problem.

Bearing these challenges in mind, we propose TFGNet, a novel target face generation network that generates HQ and identity-preserving faces from LQ inputs under textual guidance. Unlike most existing methods that rely on HQ inputs, TFGNet leverages external textual descriptions as semantic cues to guide generation from degraded images (see an example in Figure 1). To the best of our knowledge, this work is the first to introduce textual guidance for generating HQ and identity-preserving faces from degraded inputs. The significant contributions of this paper are summarized as follows:

1) We propose a unified framework for face generation that integrates a Transformer-based encoder, a codebook mechanism, and multimodal features extracted from contrastive language-image pretraining (CLIP) to produce cross-modal embeddings. These embeddings are then decoded by a diffusion model to synthesize face images with both high visual fidelity and accurate ID retention.

2) We introduce a masked diffusion loss that emphasizes identity-relevant regions and incorporate it into a total loss function with dynamically adjusted weights, promoting a balanced trade-off between visual quality and identity-consistent semantics.

3) We construct a multimodal dataset for face generation by synthesizing LQ face images and pairing them with corresponding HQ targets and manually curated textual descriptions, thus addressing the scarcity of suitable triplet data for this task.

4) We conduct extensive experiments demonstrating that the proposed TFGNet approach outperforms many state-of-the-art methods in both quantitative metrics and perceptual quality.

## 2 RELATED WORK

### 2.1 TEXT-TO-FACE GENERATION

Text-to-face generation has witnessed significant advances in recent years, which have been largely driven by the rapid development of deep learning techniques Nasir et al. (2019); Xia et al. (2021); Patashnik et al. (2021); Ding et al. (2021); Wang et al. (2021); Sun et al. (2022); Gal et al. (2022); Li et al. (2022; 2024); Kong et al. (2024); Qi et al. (2024); Li et al. (2025). GAN has been widely used for text-to-face generation due to its powerful image synthesis capabilities Nasir et al. (2019); Xia et al. (2021); Patashnik et al. (2021); Gal et al. (2022). For instance, Xia et al. (2021) proposed TediGAN, which performs text-guided face generation through StyleGAN inversion. Patashnik et al. (2021) introduced a GAN-based text-to-face generation method using CLIP-guided latent

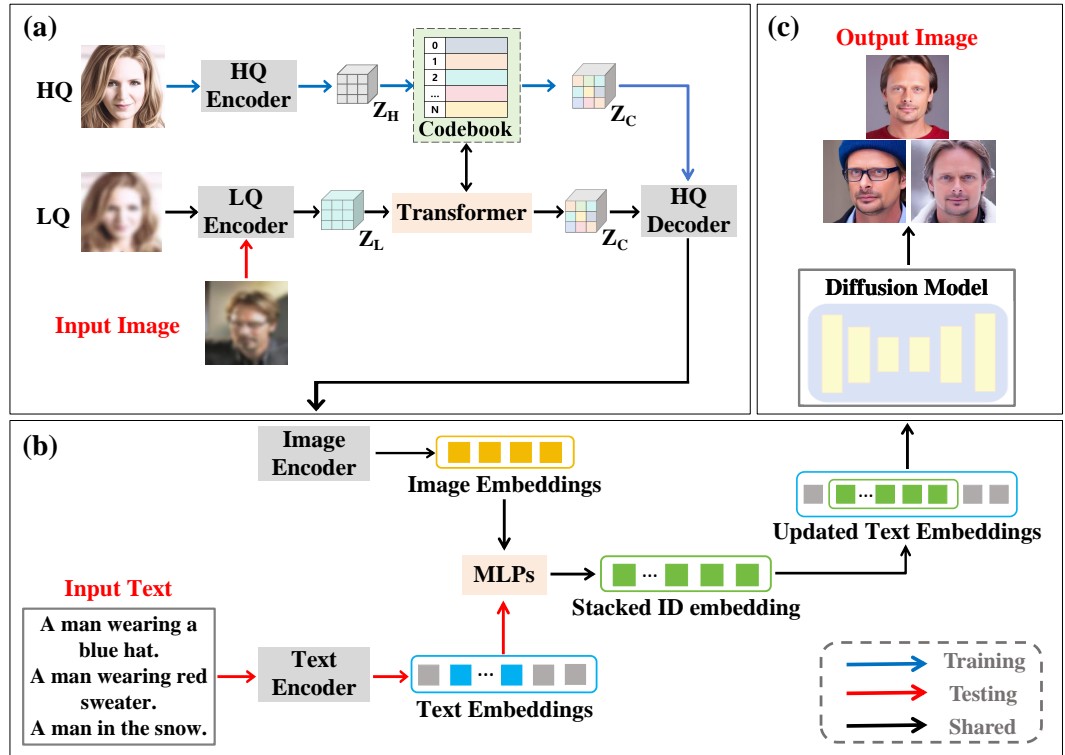

Figure 2: Overview of the proposed TFGNet architecture. (a) Image Enhancement Module: A discrete codebook and a decoder are learned via self-reconstruction on HQ face images, capturing representative visual components. A Transformer module is employed to predict the code sequence from LQ inputs, enabling the modeling of global facial structure. ($Z_H$: HQ compressed feature; $Z_L$: LQ compressed feature; $Z_C$: quantized feature.) (b) Embedding Fusion Module: Image and text embeddings are projected and merged, then concatenated along the sequence dimension to construct the stacked ID embedding. (c) Image Generation Module: The stacked ID embedding is injected into cross-attention layers within the diffusion model, facilitating adaptive identity-aware guidance during image generation.

manipulation. More recently, Transformer-based models have advanced text-to-face generation by employing self-attention mechanisms to model long-range dependencies, making them effective for capturing complex relationships between text and image modalities Ding et al. (2021); Wang et al. (2021); Sun et al. (2022); Li et al. (2022). For example, Ding et al. (2021) introduced CogView, a Transformer-based model that utilizes VQ-VAE tokenization for text-to-face synthesis. Sun et al. (2022) developed a CLIP-based Transformer framework with a two-stream architecture for flexible, free-form text-to-face generation. Diffusion models have recently emerged as a promising alternative for text-to-face generation, offering high-quality and photorealistic synthesis Li et al. (2024); Kong et al. (2024); Qi et al. (2024); Li et al. (2025). For instance, Li et al. (2024) presented an approach that incorporates stacked ID embeddings to guide the generation process. Qi et al. (2024) proposed DEADiff, which achieves diffusion-based generation by disentangling style and semantics with Q-Formers and non-reconstructive learning. Li et al. (2025) introduced PersonaMagic, a stage-regulated method with Tandem Equilibrium. Despite their impressive performance in artistic and general-purpose image synthesis, these methods typically struggle when handling degraded face images, which significantly limits their applicability in real-world scenarios.

## 3 METHODOLOGY

In this section, we propose TFGNet, a novel text-guided face generation approach that leverages textual descriptions to generate identity-preserving HQ target face images from LQ inputs. An

overview of the proposed framework is presented in Figure 2, and the details of each component within the TFGNet architecture are described below.

## 3.1 DISCRETE CODEBOOK LEARNING

The discrete codebook is learned through the self-reconstruction of HQ images. Specifically, given an HQ face image $I_h$, it is first encoded into a compressed feature representation $Z_H \in \mathbb{R}^{m \times n \times d}$ using the HQ encoder. Each feature vector in $Z_H$ is then replaced by its nearest entry in a learnable codebook, yielding a quantized feature $Z_C \in \mathbb{R}^{m \times n \times d}$ and the corresponding code token sequence. The HQ decoder subsequently reconstructs the HQ face image from $Z_C$. During training, the quantized autoencoder and codebook are jointly optimized using a combination of loss functions: the L1 loss $\mathcal{L}_1$, perceptual loss $\mathcal{L}_{per}$, and adversarial loss $\mathcal{L}_{adv}$, defined as follows:

$$\mathcal{L}_1 = \|I_h - I_{re}\|_1 , \ \mathcal{L}_{per} = \|\Phi(I_h) - \Phi(I_{re})\|_2^2 , \ \mathcal{L}_{adv} = \log D(I_h) + \log(1 - D(I_{re})), \quad (1)$$

where $I_{re}$ is the reconstructed HQ image, and $\Phi(\cdot)$ denotes the VGG19 feature extractor Simonyan & Zisserman (2014). The discriminator $D$ aims to distinguish between the real HQ image $I_h$ and the reconstructed image $I_{re}$, producing a probability score to guide adversarial training. To alleviate the under-constrained nature of image-level losses during codebook updates, an additional code-level loss $\mathcal{L}_{code}^{feat}$ is introduced to better align the encoder outputs and the quantized representations:

$$\mathcal{L}_{code}^{feat} = \|\text{sg}(Z_H) - Z_C\|_2^2 + \alpha \|Z_H - \text{sg}(Z_C)\|_2^2 , \quad (2)$$

where $\text{sg}(\cdot)$ represents the stop-gradient operator, and $\alpha = 0.5$ balances the update rates of the encoder and codebook. The overall losses for codebook prior learning $\mathcal{L}_{codebook}$ is:

$$\mathcal{L}_{codebook} = \mathcal{L}_1 + \mathcal{L}_{per} + \mathcal{L}_{adv} + \mathcal{L}_{code}^{feat} . \quad (3)$$

## 3.2 TRANSFORMER MODULE

To better capture the global facial structure from LQ inputs, a Transformer module is introduced to predict the code sequence based on the fixed codebook and HQ decoder. During training, the LQ encoder is fine-tuned to extract meaningful features from degraded inputs. Given a LQ image, it is first encoded by the fine-tuned LQ encoder to obtain a feature representation $Z_L \in \mathbb{R}^{m \times n \times d}$. This feature map is then reshaped into a sequence of $m \times n$ feature vectors $Z_L^v \in \mathbb{R}^{(m \times n) \times d}$, which is fed into the Transformer module composed of multiple self-attention blocks. The computation in the $s$-th self-attention block is defined as:

$$X_{s+1} = \text{Softmax}(Q_s K_s^T)V_s + X_s, \quad (4)$$

where $X_0 = Z_L^v$, and the query $Q_s$, key $K_s$, and value $V_s$ are obtained through linear projections of $X_s$. $T$ denotes the transpose operation. After passing through the self-attention blocks, the output is projected by a linear layer to a dimension of $(m \times n) \times C$, where $C$ is the size of the codebook. Then the Transformer predicts the code sequence $s \in \{0, \dots, C-1\}^{m \times n}$ by outputting a probability distribution over the $C$ codebook entries for each position. Based on the predicted code indices, the corresponding code vectors are retrieved from the learned codebook to form the quantized feature $Z_C \in \mathbb{R}^{m \times n \times d}$, which is subsequently decoded by the HQ decoder to generate a HQ face image. The Transformer is supervised using a combination of code-level losses. These include a cross-entropy loss $\mathcal{L}_{code}^{token}$ for code token prediction, and an L2 loss $\mathcal{L}_2$ that encourages the LQ feature to align with the corresponding quantized representation:

$$\mathcal{L}_{code}^{token} = \sum_{i=0}^{mn-1} -s_i \log(\hat{s}_i), \ \mathcal{L}_2 = \|Z_L - \text{sg}(Z_C)\|_2^2 , \quad (5)$$

where $\hat{s}_i$ is the predicted probability for the $i$-th code index. The total loss used to train the Transformer module is defined as:

$$\mathcal{L}_{transformer} = \beta \cdot \mathcal{L}_{code}^{token} + \mathcal{L}_2, \quad (6)$$

where $\beta$ is a balancing coefficient empirically set to 0.5.

### 3.3 IMAGE AND TEXT EMBEDDINGS

Our model builds upon the SDXL model (stable-diffusion-xl-base-1.0) Podell et al. (2023) and CLIP ViT-L/14 Radford et al. (2021), which serve as the foundational components of our framework. To extract semantic representations from both modalities, we employ the CLIP image encoder and text encoder to obtain image embeddings and text embeddings, respectively. To suppress identity-irrelevant information such as background and other subjects in the image, we apply a masking strategy prior to encoding. Specifically, regions outside the main body of the target ID are filled with random noise before being fed into the image encoder. This encourages the model to focus on the visual features of the intended subject.

### 3.4 STACKED ID EMBEDDING AND FUSION

To more effectively capture the ID information from input face images, we construct a Stacked ID Embedding by fusing image and text embeddings. Specifically, descriptive terms such as "man" or "woman" are first identified in the input text prompt, and their corresponding feature vectors are extracted from the text embedding $\hat{t}$. These semantic vectors are then fused with each image embedding $e^i$ using a multi-layer perceptron (MLP), resulting in a fused embedding $\hat{e}^i \in \mathbb{R}^M$. By integrating class-specific textual features, the fused embeddings provide a more expressive representation of identity-relevant information in the input images. The fused embeddings $\hat{e}^i$ from all ID images are concatenated along the sequence dimension to form the Stacked ID Embedding $s^*$:

$$s^* = \text{Concatenation}([\hat{e}^1, \cdots, \hat{e}^N]), \quad s^* \in \mathbb{R}^{N \times M}, \tag{7}$$

where $N$ is the number of ID embeddings. This representation aggregates ID cues from multiple sources while preserving the individuality of each input.

### 3.5 DIFFUSION MODEL AND GENERATION

The Stacked ID Embedding $s^*$ is incorporated into the original text embedding $\hat{t}$ to form an enhanced conditional representation. Specifically, the feature vector corresponding to a key identity-related word in $\hat{t}$ is replaced by $s^*$, resulting in an updated text embedding $t^* \in \mathbb{R}^{(L+N-1) \times M}$, where $L$ is the original text token length. The modified text embedding $t^*$ is then fed into the diffusion model for image generation. The cross-attention layers within the diffusion model support multimodal interactions and are computed as Ho et al. (2020); Nichol & Dhariwal (2021):

$$\begin{cases} Q = W_Q \cdot \phi(z_t), \ K = W_K \cdot t^*, \ V = W_V \cdot t^*, \\ \text{Attention}(Q, K, V) = \text{softmax}\left(\frac{QK^T}{\sqrt{d}}\right) \cdot V, \end{cases} \tag{8}$$

where $\phi(\cdot)$ denotes the embedding derived from the noisy latent vector $z_t$ via the UNet denoiser Ho et al. (2020), $W_Q, W_K, W_V$ are learnable projection matrices, and $d$ denotes the token dimension. This mechanism allows the model to attend to identity-specific visual features while conditioning on the full textual prompt. To further improve ID preservation and enhance the alignment between visual and textual modalities, we introduce a *masked diffusion loss* $\mathcal{L}_{masked}$. This loss guides the model to focus on semantically important regions, such as the face and clothing, while suppressing irrelevant background details. Given an input ID image $I_{id}$, we generate a binary mask $\hat{m}$ using Mask2Former Cheng et al. (2022) to localize these regions. The loss is defined as:

$$\mathcal{L}_{masked} = \mathbb{E}_{t \sim U(1,T_s)} \mathbb{E}_{I_h, I_{re}, \hat{m}} \left[ \hat{m} \cdot \| \epsilon - \epsilon_\theta(I_t, t, s^*, \hat{m}) \|^2 \right], \tag{9}$$

where $\epsilon$ is the ground-truth noise, $\epsilon_\theta$ is the predicted noise by the diffusion model, $T_s$ is the total number of steps, and $I_t$ is the noisy image at time step $t$. The element-wise multiplication with $\hat{m}$ ensures that the supervision signal is concentrated on identity-relevant areas. This design enables the model to generate HQ identity-consistent images while mitigating artifacts from irrelevant regions.

### 3.6 JOINT TRAINING

The framework is optimized in two stages. In the first stage, discrete codebook learning is performed through HQ image self-reconstruction, enabling the codebook to acquire compact visual tokens while the Transformer models their dependencies for reliable reconstruction. In the second stage,

the entire model is trained jointly on the same dataset, with the text encoder frozen to preserve the pre-trained semantic space. Since explicit text annotations are unavailable in the training dataset, we leverage CLIP's image-text alignment to construct *pseudo-text representations*. Specifically, each training image is first encoded by the CLIP image encoder to obtain a high-dimensional semantic embedding in the joint vision-language space. To enhance expressiveness and reduce the modality gap, this embedding is further projected through a lightweight MLP into the text embedding space. The resulting vectors act as pseudo-text tokens, which are then combined with image embeddings to form identity-aware conditions, guiding the model to synthesize HQ images with consistent identity, high visual fidelity, and semantic alignment. The overall objective is formulated as:

$$\mathcal{L}_{total} = \lambda_1 \cdot \mathcal{L}_{codebook} + \lambda_2 \cdot \mathcal{L}_{transformer} + \lambda_3 \cdot \mathcal{L}_{masked}, \tag{10}$$

where $\lambda_1, \lambda_2, \lambda_3$ are empirically tuned to balance the contributions of each component. The Adam optimizer Kingma & Ba (2015), combined with stage-wise learning rate scheduling, is employed to stabilize convergence and improve performance, enabling the final model to synthesize high-quality, identity-consistent, and semantically faithful images.

## 4 EXPERIMENTS

### 4.1 DATASETS AND EVALUATION METRICS

**Training Dataset.** We use the FFHQ dataset[1], which contains 70,000 HQ face images collected from Flickr, each with a resolution of $1024 \times 1024$. The dataset exhibits diverse variations in age, gender, ethnicity, facial expression, head pose, background, and lighting conditions. We randomly select 10,000 images from FFHQ and resize them to $512 \times 512$ as the training set. The following experiments demonstrate that our method achieves strong performance using only a small subset of the dataset, thus avoiding reliance on all 70,000 images and reducing computational costs. This subset is used to train the image-related modules of our framework. Since the CLIP text encoder has been pre-trained on a large-scale text-image corpus covering a wide range of semantic labels, it is used without additional fine-tuning. All experiments are conducted on an NVIDIA GeForce RTX 4090 D GPU.

**Testing Dataset.** We evaluate the performance of all competing methods on two test datasets. The first is derived from a competition dataset Face500, which comprises 500 subfolders, each containing a severely degraded face image, a target HQ image, and a textual description. To further assess the generalization ability of our approach, we construct an additional test set based on the publicly available CelebA dataset Liu et al. (2015). Specifically, we randomly select 500 image pairs from CelebA (each consisting of two images of the same ID, i.e., the same person), apply the same degradation process used in the Face500 dataset to one image in each pair, and manually create textual descriptions reflecting the ID and visual attributes of the corresponding face. Note that the description needs to include a gender-related keyword to prevent possible mistakes. We refer to this dataset as T2F500, which addresses the lack of publicly available benchmarks that provide aligned triplets of degraded images, textual descriptions, and HQ targets, thereby enabling more comprehensive and rigorous evaluation of text-guided face generation methods.

**Quantitative Metrics.** To quantitatively evaluate the performance of our method, we adopt several widely used evaluation metrics, each grounded in established pre-trained models. *Face Similarity (Face Sim.)* is computed using ArcFace Deng et al. (2019) to assess ID preservation by comparing facial feature embeddings between the generated and ground-truth images. *DINO* Caron et al. (2021) is employed to evaluate visual similarity based on high-level semantic representations. *CLIP-I* and *CLIP-T*, both derived from the CLIP-ViT-B/32 model Radford et al. (2021), are used to measure image-image consistency and text-image alignment, respectively. *LPIPS* Zhang et al. (2018) quantifies perceptual similarity by comparing deep feature distances extracted using a pre-trained AlexNet. *FID* Heusel et al. (2017) evaluates overall image quality by computing the Fréchet distance between the distributions of real and generated images in a feature space.

---

[1]https://github.com/NVlabs/ffhq-dataset

Table 1: Comparisons on the T2F500 dataset.

| | Face Sim.↑ (%) | DINO↑ (%) | CLIP-I↑ (%) | CLIP-T↑ (%) | LPIPs↓ | FID↓ |
|---|---|---|---|---|---|---|
| TediGAN Xia et al. (2021) | 13.45 | 38.13 | 60.55 | **36.81** | 0.7554 | 171.31 |
| ELITE Wei et al. (2023) | 0.48 | 3.36 | 48.30 | 16.11 | 0.8780 | 428.45 |
| SSR_Encoder Zhang et al. (2023) | 10.79 | 23.90 | 55.66 | 17.68 | 0.8528 | 197.41 |
| CelebBasis Yuan et al. (2023) | 20.16 | 54.59 | 63.90 | 31.25 | 0.7825 | 94.7 |
| HyperDreamBooth Ruiz et al. (2023) | 19.16 | 39.39 | 65.82 | 22.03 | 0.7769 | 127.09 |
| FreeCustom Ding et al. (2024) | 21.61 | 37.15 | 64.28 | 24.09 | 0.8115 | 138.94 |
| TheChosenOne Avrahami et al. (2024) | 16.73 | 42.43 | 63.46 | 25.20 | 0.7616 | 129.01 |
| MuDI Jang et al. (2024) | 17.14 | 40.55 | 64.30 | 23.35 | 0.7822 | 130.51 |
| FastComposer Xiao et al. (2024) | 19.49 | 26.67 | 52.81 | 19.45 | 0.7725 | 186.16 |
| MasterWeaver Wei et al. (2024) | 21.85 | 49.81 | 62.91 | 31.00 | 0.7751 | 112.45 |
| InstantID Wang et al. (2024) | **27.79** | 53.30 | **71.70** | 23.85 | 0.7719 | 108.34 |
| PuLID Guo et al. (2024) | 25.04 | 43.47 | 65.73 | 27.66 | 0.7933 | 166.09 |
| MoMA Song et al. (2024) | 26.03 | 59.49 | 68.91 | 31.44 | 0.7132 | 89.56 |
| ConsistentID Huang et al. (2024) | 23.16 | 46.87 | 69.35 | 27.06 | 0.7713 | 131.83 |
| DEADiff Qi et al. (2024) | 25.77 | 54.72 | 65.70 | 29.54 | 0.7590 | 113.22 |
| Photomaker Li et al. (2024) | 27.01 | **62.12** | 69.90 | **32.75** | **0.7046** | **80.04** |
| PersonaMagic Li et al. (2025) | 22.22 | 58.00 | 62.57 | 31.16 | 0.7511 | 109.37 |
| **TFGNet (Ours)** | **32.50** | **63.07** | **73.23** | 31.23 | **0.6966** | **76.77** |

Table 2: Comparisons on the Face500 dataset.

| | Face Sim.↑ (%) | DINO↑ (%) | CLIP-I↑ (%) | CLIP-T↑ (%) | LPIPs↓ | FID↓ |
|---|---|---|---|---|---|---|
| TediGAN Xia et al. (2021) | 15.94 | 37.60 | 55.10 | **33.80** | 0.7732 | 172.38 |
| ELITE Wei et al. (2023) | 0.51 | 2.55 | 43.67 | 21.23 | 0.9056 | 432.59 |
| SSR_Encoder Zhang et al. (2023) | 14.59 | 16.38 | 52.39 | 21.82 | 0.8624 | 238.86 |
| CelebBasis Yuan et al. (2023) | 19.44 | 50.72 | 53.82 | 29.74 | 0.7988 | 101.37 |
| HyperDreamBooth Ruiz et al. (2023) | 22.98 | 45.02 | 63.21 | 24.76 | 0.7953 | 102.46 |
| FreeCustom Ding et al. (2024) | 4.72 | 14.29 | 54.17 | 24.92 | 0.8469 | 270.32 |
| TheChosenOne Avrahami et al. (2024) | 15.00 | 27.60 | 56.60 | 26.09 | 0.7938 | 194.13 |
| MuDI Jang et al. (2024) | 19.97 | 32.05 | 60.12 | 25.44 | 0.8216 | 165.28 |
| FastComposer Xiao et al. (2024) | 20.49 | 22.92 | 46.92 | 21.11 | 0.7963 | 214.80 |
| MasterWeaver Wei et al. (2024) | 26.17 | 41.95 | 54.98 | 28.50 | 0.7775 | 158.30 |
| InstantID Wang et al. (2024) | 26.85 | 54.51 | 67.60 | 28.16 | 0.7492 | **77.29** |
| PuLID Guo et al. (2024) | 28.63 | 28.43 | 59.12 | 26.72 | 0.8628 | 203.85 |
| MoMA Song et al. (2024) | 30.19 | 57.14 | 64.86 | **30.62** | 0.7443 | 92.54 |
| ConsistentID Huang et al. (2024) | 33.43 | 36.32 | 64.85 | 25.82 | 0.8205 | 175.37 |
| DEADiff Qi et al. (2024) | 33.75 | 48.70 | 63.55 | 28.19 | 0.8040 | 137.27 |
| Photomaker Li et al. (2024) | **39.51** | **59.05** | **71.93** | 29.65 | **0.7193** | 82.81 |
| PersonaMagic Li et al. (2025) | 16.62 | 51.48 | 53.22 | 28.60 | 0.8125 | 140.46 |
| **TFGNet (Ours)** | **44.80** | **62.04** | **78.04** | 29.60 | **0.7152** | **79.66** |

## 4.2 COMPARISONS WITH STATE-OF-THE-ART METHODS

We compare the proposed TFGNet[2] with many state-of-the-art methods under consistent experimental settings, *i.e.*, by using the same input images and corresponding text prompts. Due to the limited space, more results can be found in the supplementary material. Quantitative results are presented in Table 1 and Table 2, red and blue indicate the best and the second performance. TFGNet consistently outperforms competing approaches across key evaluation metrics, including Face Sim., DINO, CLIP-I, LPIPS, and FID, demonstrating a strong capacity for ID preservation and facial detail reconstruction, both of which are essential for HQ face generation. For the CLIP-T metric, which measures text-image alignment, TFGNet prioritizes visual fidelity and ID consistency, treating the textual prompt as auxiliary guidance. This design choice may result in slightly lower CLIP-T scores, which are expected and do not undermine the overall performance. For visual comparison, we consider five state-of-the-art methods in identity-aware face generation: MoMA Song et al. (2024), ConsistentID Huang et al. (2024), DEADiff Qi et al. (2024), Photomaker Li et al. (2024) and Per-

---

[2]More results can be seen in Appendix.

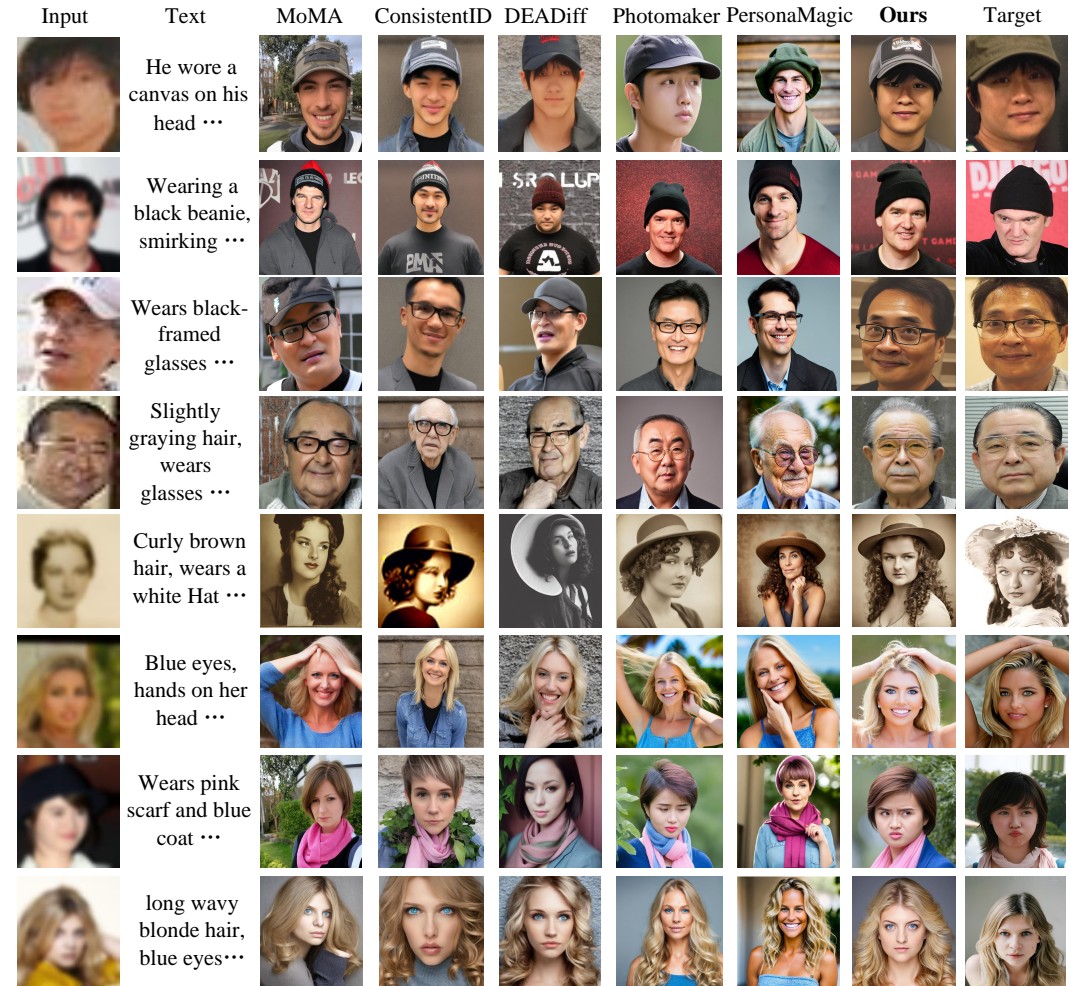

Figure 3: Visual comparisons with five state-of-the-art methods for face generation.

sonaMagic Li et al. (2025). TFGNet achieves superior identity preservation and more faithful detail reconstruction. As illustrated in Figure 3, TFGNet achieves superior visual alignment with ground-truth targets, outperforming other methods in terms of perceptual quality and structural coherence. In terms of computational efficiency, TFGNet achieves a favorable balance between inference speed and generation accuracy. Although it introduces a moderate computational overhead, the resulting improvements in image quality justify this trade-off when compared to faster but less accurate methods. In summary, TFGNet establishes a new benchmark in both quantitative metrics and visual quality, highlighting its effectiveness and robustness for text-guided face generation tasks.

## 4.3 ABLATION STUDIES

To comprehensively evaluate the contribution of individual components in TFGNet, we conducted ablation experiments by removing specific modules while keeping other settings unchanged. The results are summarized in Table 3.

**Codebook.** The codebook constrains encoder outputs to a discrete set of vectors, thereby stabilizing the latent space and enabling consistent reconstruction of low-level facial features, including edges, contours, and textures. When this module is removed, features may drift during training, causing inconsistencies or distortions. Consequently, quantization acts as a fundamental prior, providing stable feature representation and facilitating subsequent processing stages.

Table 3: Ablation study on different components of TFGNet. "✓" indicates the module is enabled, and "✗" indicates the module is disabled.

| Modules | | | | Metrics | | |
|---|---|---|---|---|---|---|
| Codebook | Transformer | Stacked ID Embedding | Masked Diffusion Loss | Face Sim.↑(%) | LPIPS↓ | FID↓ |
| ✗ | ✓ | ✓ | ✓ | 26.60 | 0.7014 | 77.11 |
| ✓ | ✗ | ✓ | ✓ | 30.04 | 0.7039 | 81.97 |
| ✓ | ✓ | ✗ | ✓ | 22.28 | 0.7388 | 96.98 |
| ✓ | ✓ | ✓ | ✗ | 24.36 | 0.7029 | 80.78 |
| ✓ | ✓ | ✓ | ✓ | **32.50** | **0.6966** | **76.77** |

**Transformer.** The Transformer encodes long-range dependencies and global context across facial regions, ensuring structural consistency in the generated faces. When this module is removed, the network relies solely on local feature interactions, which can produce misaligned or fragmented structures. Consequently, global attention is essential for generating faces that are detailed locally while maintaining global coherence, preserving harmonious relationships among facial components such as the eyes, nose, and mouth.

**Stacked ID Embedding.** Stacked ID embedding preserves personalized ID across generated samples. By integrating multiple ID cues, it maintains subtle yet distinctive traits while following textual guidance. When this module is removed, the network relies only on high-level textual descriptions, capturing general attributes but losing fine-grained ID details. Thus, explicit ID embedding is crucial for both personalized and semantically controlled face generation.

**Masked Diffusion Loss.** Masked diffusion loss focuses on learning regions sensitive to ID, such as the eyes, nose, and mouth, which are crucial for preserving personal characteristics. When this module is removed, supervision spreads uniformly, weakening the signal and reducing fidelity in key features. Thus, a carefully designed loss ensures robust ID preservation and high-quality face generation.

Each component of TFGNet contributes uniquely. The codebook stabilizes low-level features, the Transformer ensures global coherence, the stacked ID embedding preserves ID, and the masked diffusion loss supervises key regions. Removing any module degrades performance, confirming the necessity of the full design.

## 5 CONCLUSION

In this paper, we presented TFGNet, a novel target face generation network from LQ inputs via textual guidance. Unlike most existing methods that depend on HQ inputs, TFGNet leverages external textual descriptions as semantic guidance to directly generate HQ and identity-preserving faces from degraded images, significantly enhancing generation performance under complex degradation conditions. It integrated a Transformer-based encoder, a discrete codebook mechanism, and CLIP-derived multimodal features to produce cross-modal embeddings, which were subsequently decoded by a diffusion model to generate HQ face images. To further improve generation fidelity, we introduced a masked diffusion loss focused on identity-relevant regions and embedded it into a dynamically weighted total loss function, enabling a balanced trade-off among visual fidelity, semantic alignment, and ID preservation. In addition, we constructed a multimodal dataset comprising synthetically LQ face images, their HQ target counterparts, and manually curated textual descriptions, addressing the scarcity of suitable triplet data for this task. Extensive experimental results demonstrated that TFGNet outperforms many state-of-the-art methods across multiple quantitative metrics and in terms of perceptual quality.

## 6 USE OF LLMs

During the preparation of this paper, we used a large language model solely for writing assistance and language polishing. It was not involved in the conception of ideas, the design of methods, the execution of experiments, or the interpretation of results.

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

# A APPENDIX

## A.1 DETAILS OF DATASET IMAGE DEGRADATION PREPROCESSING

All HQ images used in the datasets have a fixed resolution of 512×512. A unified degradation pipeline is employed for all processes of generating degraded images in the construction of both the training and testing datasets. First, HQ images are convolved with a Gaussian blur kernel, whose standard deviation is randomly sampled from the range [1, 10]. Second, the images are downsampled by a factor randomly chosen from [1, 10] to reduce resolution. Third, Gaussian noise with a standard deviation sampled from [1, 15] is added to simulate noise interference. Then, JPEG compression is applied with a quality factor randomly selected from [30, 90] to reflect different levels of compression artifacts. Finally, the degraded images are resized back to a fixed resolution of 128×128 as the LQ images.

## A.2 RESULTS UNDER NORMAL VIEW

Figure 4 presents more results of TFGNet under the normal view to demonstrate its effectiveness. It can be observed that the generated images are closer to the ground truth, with clearer details especially in hair textures and clothing features.

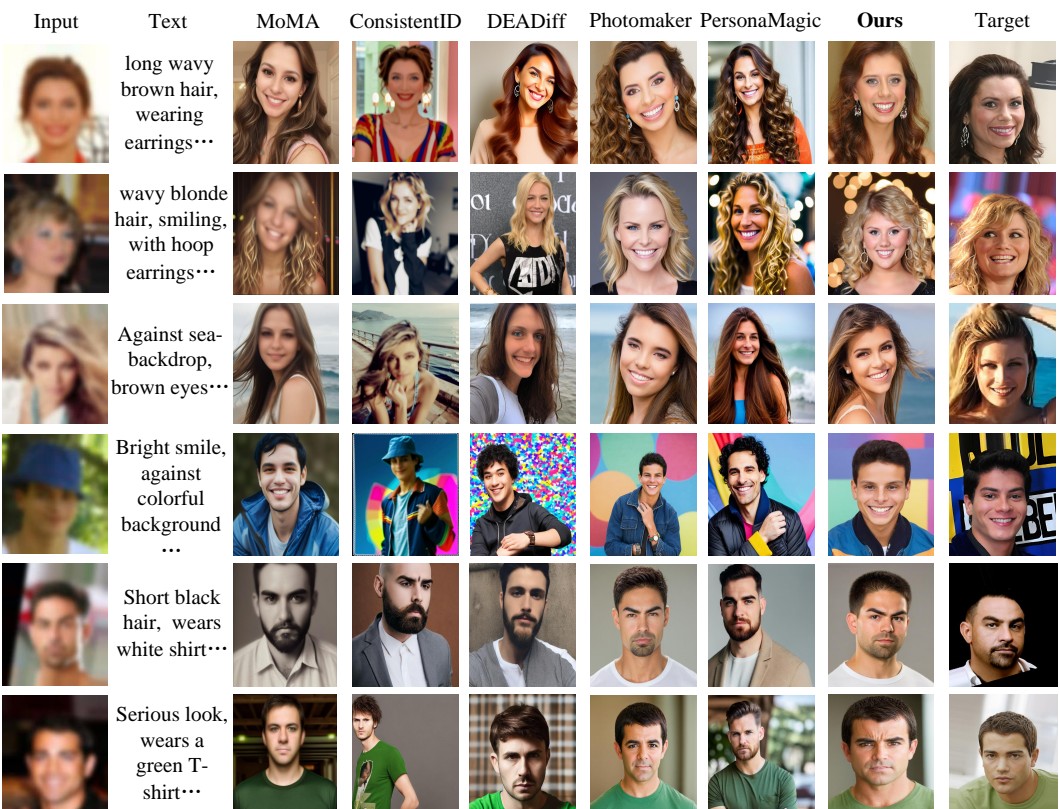

Figure 4: Additional results under Normal View: MoMA Song et al. (2024), ConsistentID Huang et al. (2024), DEADiff Qi et al. (2024), Photomaker Li et al. (2024) and PersonaMagic Li et al. (2025).

## A.3 RESULTS UNDER COMPLEX SITUATIONS

Situations encountered in social security are complex and diverse. Many of the face images captured by surveillance are in profile view and suffer from occlusions caused by clothing accessories such as hats or glasses. However, our work is capable of transforming these images into frontal views,

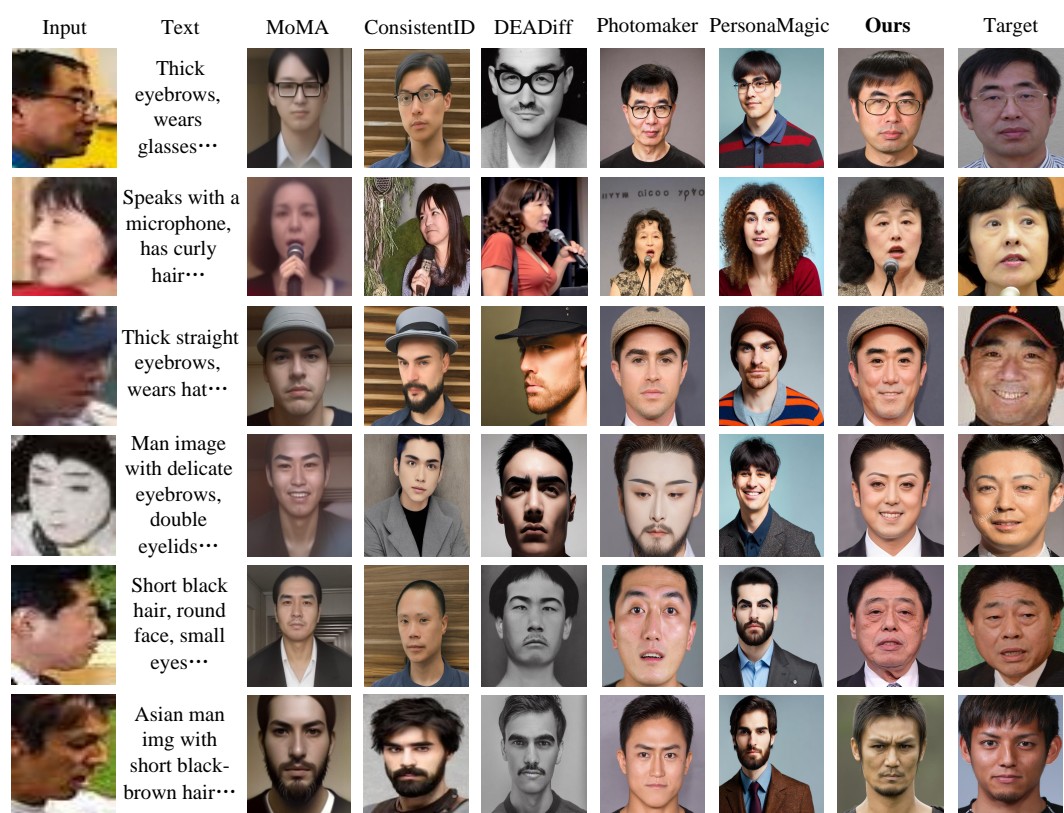

Figure 5: Additional results under complex situations: MoMA Song et al. (2024), ConsistentID Huang et al. (2024), DEADiff Qi et al. (2024), Photomaker Li et al. (2024) and PersonaMagic Li et al. (2025).

which is significantly beneficial for ID recognition. The results are shown in Figure 5, and we can specifically construct such a dataset in future work.

### A.4 VISUAL RESULTS OF OTHER COMPARISON METHODS

In this section, we present the visual results of several additional comparison methods. The results are shown in Figure 6 and 7. For the remaining methods mentioned in the main text but not shown here, some were excluded due to the generation of sensitive images that are inappropriate for publication. Others were omitted because they are not specifically designed for text-to-image generation of human faces and lack a face recognition module, resulting in outputs that barely resemble human figures.

### A.5 REPRODUCIBILITY STATEMENT

The code and datasets will be made available upon acceptance of the paper, ensuring all key experiments can be replicated by the research community.

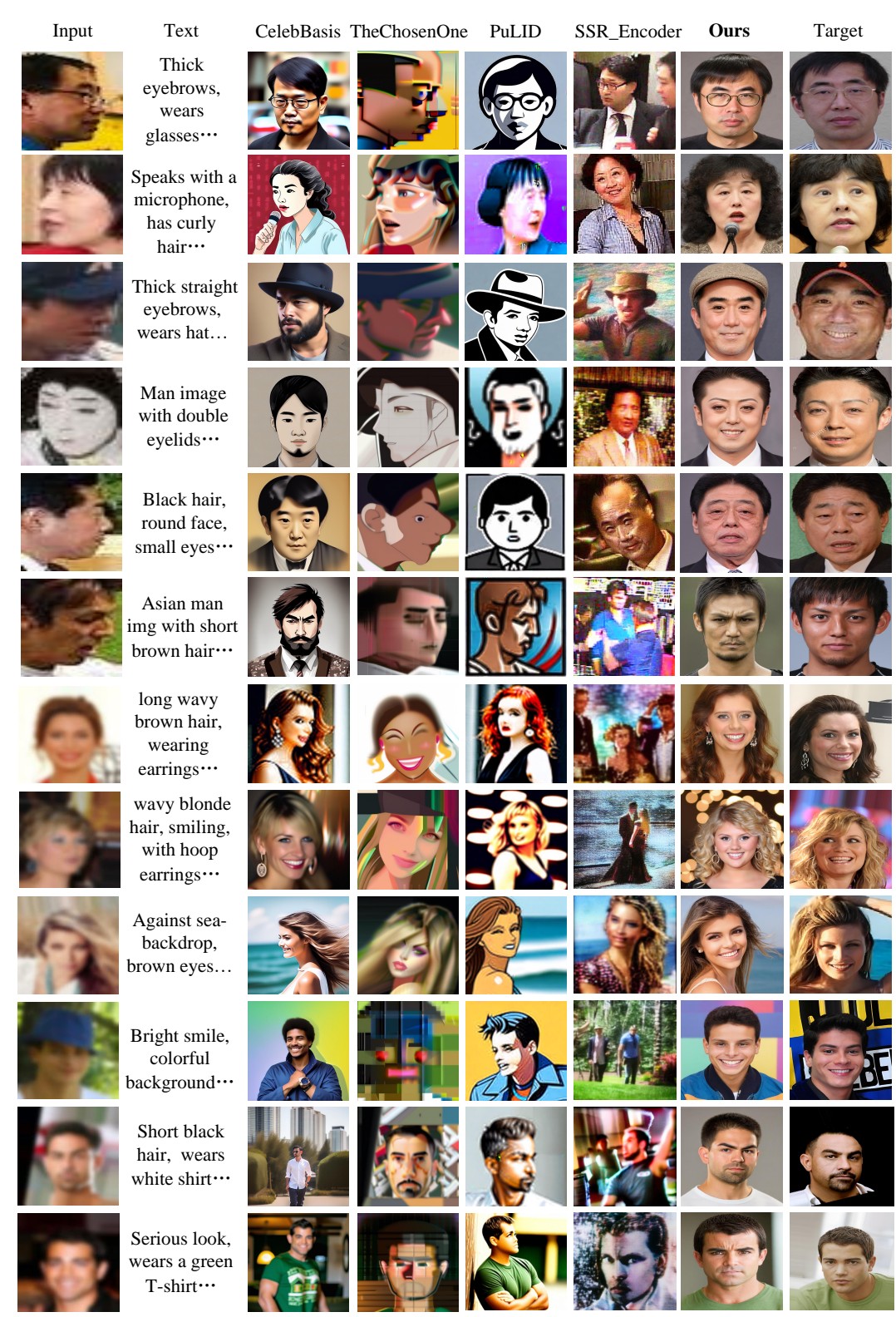

Figure 6: Visual comparisons with other methods: CelebBasis Yuan et al. (2023), TheChosenOne Avrahami et al. (2024), PuLID Guo et al. (2024), SSR_Encoder Zhang et al. (2023).

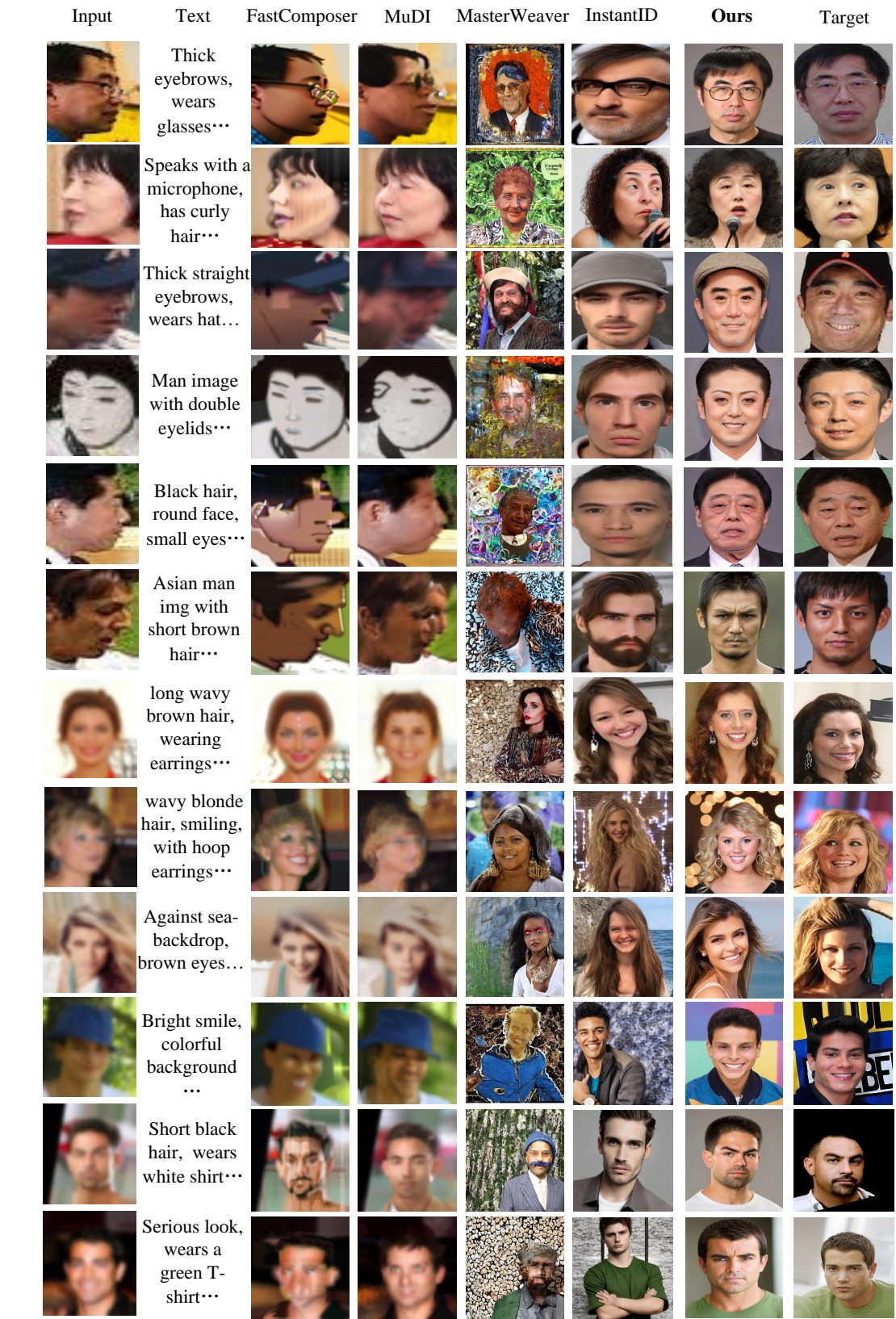

Figure 7: Visual comparisons with other methods: FastComposer Xiao et al. (2024), MuDI Jang et al. (2024), MasterWeaver Wei et al. (2024), InstantID Wang et al. (2024).

