# OpenReview forum: "TFGNet: Target Face Generation from Low-Quality Images via Textual Guidance"
_ICLR.cc/2026/Conference — Submitted to ICLR 2026_

### Official Review · Reviewer_wHno · 2025-10-21

**Soundness:** 2
**Presentation:** 2
**Contribution:** 2
**Rating:** 6
**Confidence:** 2

**Summary:**

This paper introduces TFGNet, a novel framework for generating high-quality, identity-preserving faces from a single low-quality image and a textual description. The paper proposes a masked diffusion loss to strike a balanced trade-off between visual quality and identity-consistent semantics. The paper also proposes a dataset for the task.

**Strengths:**

1. The paper addresses the important problem of restoring high-fidelity, identifiable faces from low-quality images.
2. The paper proposes valuable benchmark T2F500.

**Weaknesses:**

1. The paper does not clarify whether the text prompt is used for restoration (filling in lost details lost) or for editing (incorporating new attributes).
2. What happens if the text prompt contradicts the HQ target's attributes?
2. There are no ethical considerations mentioned in the paper.

**Questions:**

What happens if the text prompt contradicts the HQ target's attributes?

---

> ### Author Response · Authors · 2025-11-17
>
> Thank you for your thorough review and valuable comments on our manuscript.
>
> Regarding the purpose of text prompts, we apologize for the lack of clarity in our original presentation. The core objective of TFGNet is text-guided face image editing, which involves modifying facial attributes based on textual descriptions while maintaining identity consistency. This differs fundamentally from traditional image restoration, as our approach focuses on achieving controllable facial attribute transformations through textual instructions rather than simply enhancing image quality. We have now clarified this key aspect in the methodology section of the paper.
>
> Concerning the potential conflict between text prompts and identity attributes, this challenge is effectively addressed through our proposed masked diffusion loss mechanism. When textual descriptions conflict with identity characteristics, the model prioritizes identity preservation. Specifically, the identity preservation module provides strong feature constraints, while the masked diffusion loss focuses on editable regions to minimize interference with critical identity-related areas at the pixel level. This design enables the model to fulfill editing requirements while maintaining identity features to the greatest extent possible.
>
> Ethical Considerations: We fully acknowledge the importance of ethical considerations. All datasets used in our work are publicly available face datasets obtained from legitimate sources, and no private or sensitive data are used. Our method is intended solely for research and other lawful applications, and we do not support or encourage any misuse.
>
> Thank you again for your insightful comments, which have significantly improved the quality of our paper. We hope our responses meet with your approval.

---

> ### Author Response · Authors · 2025-11-29
> **Summary During the Discussion Period**
>
> Dear Area Chair, here are the discussion records between us and the reviewer for your kind attention. We summarize below the key concerns and questions raised by the reviewer and our responses.
>
> Reviewer wHno:
>
> The reviewer acknowledged that TFGNet addresses a valuable problem of LQ→HQ text-guided face generation, particularly enabling controllable facial attribute editing while preserving identity. The reviewer also recognized the proposed T2F500 benchmark and the importance of restoring high-fidelity, identifiable faces.
>
> However, the reviewer raised the following main issues:
>
> 1. Clarification on the purpose of text prompts: Is the text used for image restoration (filling lost details) or for editing (adding new attributes)?
>
> 2. Conflict between text prompts and HQ target attributes: What happens if the textual description conflicts with the identity attributes of the HQ image?
>
> 3. Ethical considerations: The paper did not address potential ethical issues.
>
> Our Response:
>
> Response to the three issues raised by the reviewer is summarized as follows:
>
> 1. Purpose of text prompts
>
> The core objective of TFGNet is text-guided face image editing, focusing on controllably modifying facial attributes based on textual descriptions while maintaining identity consistency. This is fundamentally different from traditional image restoration, which only aims to enhance image quality.
>
> 2. Conflict between text prompts and identity attributes
>
> When textual descriptions conflict with identity features, the model prioritizes identity preservation. The identity preservation module provides strong feature constraints, and the masked diffusion loss focuses on editable regions, minimizing interference with critical identity-related areas, thus balancing text-guided edits with identity fidelity.
>
> 3. Ethical considerations
>
> We fully acknowledge the importance of ethical considerations. All datasets are publicly available and legally obtained, with no private or sensitive information used. The method is intended solely for research and other lawful applications, and misuse is neither supported nor encouraged.

---

### Official Review · Reviewer_5ejV · 2025-10-23

**Soundness:** 2
**Presentation:** 3
**Contribution:** 2
**Rating:** 2
**Confidence:** 4

**Summary:**

This work targets the task of text-guided face image generation with low-quality reference, aiming to combine identity information from the low-quality input with specific descriptions from the text prompt for human face image synthesis. Specifically, the authors employ a codebook-based VQGAN-like model to restore the low-quality input image, then integrate the restored image features with text prompt features into stacked identity embeddings for diffusion-based generation. A masked diffusion loss is further proposed to balance identity fidelity and visual quality, and a curated dataset aligned with this task - pairing low-quality images with corresponding text prompts as inputs - is also proposed.

**Strengths:**

This work presents a relatively unexplored task of text-guided face generation from low-quality reference images, conducting an exploration in this specific area. The proposed model demonstrates the capability to produce results with relatively high visual quality that align with the provided text prompts. Furthermore, it achieves superior performance compared to existing baseline methods across the evaluation metrics established by the authors.

**Weaknesses:**

1. The task setting possesses inherent ambiguity - a single low-quality input can correspond to multiple plausible high-fidelity ground truths, which complicates a definitive evaluation and challenges the clear validation of the method's superiority.

2. The proposed pipeline can be essentially decomposed into the straightforward sequential combination of two well-established sub-tasks: face image restoration and text and reference-guided face generation, both of which have been extensively studied. Consequently, the methodological design, which directly cascades two models as mentioned above, lacks integration and novelty.

3. For specific model design, the initial restoration stage closely resembles existing Vector Quantization-based models like VQGAN [1], while the subsequent stacked identity embedding strategy bears strong similarity to the approach introduced in PhotoMaker [2].


[1] Esser, Patrick, Robin Rombach, and Bjorn Ommer. "Taming transformers for high-resolution image synthesis." Proceedings of the IEEE/CVF conference on computer vision and pattern recognition. 2021.

[2] Li, Zhen, et al. "Photomaker: Customizing realistic human photos via stacked id embedding." Proceedings of the IEEE/CVF conference on computer vision and pattern recognition. 2024.

**Questions:**

1. Please elaborate on the practical significance and value of the proposed setting in this work, and clearly distinguish its contributions from a straightforward, sequential pipeline that simply combines "face image restoration" followed by "text/reference-based image generation."

2. Please discuss the specific distinctions and potential superiority between the proposed restoration model and existing Vector Quantization-based generative models, as well as the advantages of the stacked ID embedding used in the second stage compared to the one proposed in PhotoMaker.

3. Please justify the rationale behind using ArcFace feature similarity between the generated images and the ground-truth images for evaluating model performance in this specific setting, considering the inherent ambiguity of the task.

4. The article mentions the contribution of a multimodal dataset. However, detailed descriptions are lacking. Please introduce this dataset and discuss its specific value or advantages compared to existing Text-to-Image datasets.

---

> ### Author Response · Authors · 2025-11-17
>
> We sincerely thank you for your thoughtful feedback and constructive comments.
> Response to Weaknesses and Question 1 & 2:
> We would like to emphasize that the primary contribution of our work is the first explicit proposal and definition of a new and practically important task: generating high-quality (HQ) identity images from a single low-quality face image and a textual description.
>
> The core challenge of this task lies in its inherent one-to-many mapping nature, which is not a drawback but rather the key to its real-world relevance. A representative example is criminal investigation, where a suspect’s portrait must be reconstructed from surveillance footage together with eyewitness descriptions. In such scenarios, the conventional “restore-then-generate” pipeline fails: restoration models arbitrarily hallucinate one possible clean face, losing all other plausible identities, while generation models alone lack reliable identity cues extracted from the LQ image.
>
> For this reason, TFGNet is designed specifically to address this new task. Inspired by the workflow of police composite artists, we integrate the two subtasks into a unified model and optimize them jointly. This allows the system to effectively navigate the large space of uncertainties, balancing identity evidence from the LQ input with creative variations guided by text.
>
> We are the first to recognize that these two subtasks must be solved collaboratively within a unified framework, which is crucial for pushing this line of research toward real-world deployment.
>
> Response to Question 3：
> We thank the reviewer for this important question. Our use of ArcFace is based on the following rationale:
>
> The core challenge of our task is to balance identity preservation against text adherence. While visual details (e.g., pose, lighting) are uncertain, the identity itself is determinate, originating from the single low-quality input image.
>
> ArcFace serves as an "identity anchor", rigorously evaluating whether the model deviates from the input identity when following textual guidance. A high ArcFace score indicates a reliable identity preservation capability, which is fundamental to the task.
>
> To comprehensively capture model performance across different dimensions, we introduced a suite of complementary metrics: CLIP-T directly quantifies text-image alignment, FID and LPIPS jointly assess the overall visual quality and perceptual similarity of the outputs, while DINO and CLIP-I focus on measuring high-level semantic consistency between images. This comprehensive evaluation protocol ensures that we can rigorously constrain identity fidelity while objectively assessing the model's performance in semantic adherence and visual quality.
>
> Response to Question 4：
> We thank the reviewer for the interest in our dataset. The core innovation and value of our constructed T2F500 dataset lie in its provision of the first identity-consistent yet visually independent triplets of (LQ image, Text, HQ image) for the task of text-guided identity-preserving generation.
>
> Crucially, unlike face super-resolution datasets that provide only LQ-HQ pairs, our HQ target image shares the identity with the LQ input but can differ significantly in visual attributes such as pose and expression. Concurrently, the text description accurately depicts the visual content of this specific HQ target image.
>
> This design fundamentally defines our task: it requires the model to go beyond simple "super-resolution restoration" and learn to disentangle identity from attributes—it must leverage the LQ input to anchor the identity, and then synthesize a entirely new and plausible HQ image based on the text description (whose semantics align with the HQ target). This perfectly simulates real-world applications like generating a suspect composite from a blurry surveillance feed and an eyewitness account.
>
> Therefore, our dataset fills a critical gap. It is not a traditional "restoration" dataset but the first benchmark established for the new paradigm of "Text-guided Identity-preserving Image Generation". It not only facilitates the training of models to balance identity preservation with semantic control but also provides an indispensable platform for the fair evaluation of model performance on this challenging task.

---

> > ### Comment · Reviewer_5ejV · 2025-11-24
> > **Reviewer Response to Author Rebuttal**
> >
> > First, I would like to thank the authors for their explanations regarding the necessity and distinctiveness of the task. The clarifications provide a reasonable context for this work. However, I still maintain that the methodological novelty is limited. Furthermore, regarding the evaluation, I find the use of ArcFace to measure the similarity between the model output and a specific identity to be unreasonable in this scenario. Since the input distorted images have lost their identity information, the problem is inherently ill-posed, meaning that multiple different identities could be considered reasonable results. Therefore, strictly evaluating identity consistency against a single designated identity does not align with the nature of this task. This implies that the model achieving the highest score on this metric is not necessarily the best performing one.

---

> ### Author Response · Authors · 2025-11-26
>
> We sincerely thank the reviewer for the valuable and thoughtful comments. Your feedback has helped us further refine the clarity and positioning of our work.
>
> We would like to respectfully clarify an important point about the ArcFace evaluation. In our study, ArcFace is not used to compare the generated result with the low quality input image. Instead, it is used exclusively to measure the identity similarity between the generated high quality image and the predefined high quality ground truth target. Since the LQ image lacks sufficient and reliable identity details, it is not suitable for identity similarity computation and therefore is not involved in any ArcFace evaluation.
>
> This design is fully consistent with the objective of TFGNet. The goal of our method is to reconstruct an identity-faithful high quality face image under the joint guidance of a degraded input and a textual description. Such a setting corresponds to practical application scenarios, for example in security or forensic assistance, where only a low quality surveillance frame and a witness description are available, yet a high quality identity-consistent reconstruction is required.
>
> As illustrated in Figure 3 of the paper, the rightmost Target column is the ground truth HQ identity image, and all identity consistency metrics are calculated relative to this Target rather than the LQ input.
>
> We hope this clarification and contextual explanation address the misunderstanding. We greatly appreciate the reviewer’s careful reading and constructive feedback.

---

> > ### Comment · Reviewer_5ejV · 2025-11-26
> > **Reviewer Response**
> >
> > Thank you for the authors' response. Yet, in my previous discussion above, I was also referring to the fact that it is not reasonable enough to "compute the ArcFace feature similarity between the model output and the high-resolution (HR) image corresponding to the low-resolution (LR) image". This is because, as also pointed out by the authors, the LR image loses identity information, which leads to multiple plausible HR solutions corresponding to it.

---

> ### Author Response · Authors · 2025-11-28
>
> Thank you very much for your continued attention to the soundness of our evaluation protocol. Your questions have helped us articulate more clearly the relationship among our dataset design, task objectives, and evaluation methodology. We apologize for not understanding your concerns earlier.
>
> Allow us to clarify the construction logic of the “same-ID matching” strategy used in our test datasets. Our previous description may not have been sufficiently clear, and we have now added an explanatory sentence to address this.
>
> In our test datasets, each pair of “LQ input” and “HR target” strictly comes from two different high-quality images of the same identity (ID). Specifically, we first select a clean HR image of a fixed ID as the HR target, ensuring that the identity reference for evaluation is well-defined.
>
> Then, we take another HR image of the same ID and apply a unified degradation pipeline—including Gaussian blur, random downsampling, Gaussian noise, and JPEG compression (details provided in Appendix A.1)—to generate the corresponding LQ input.
>
> Meanwhile, we write an ID-specific textual description based on that subject’s unique visual attributes (such as facial features, hairstyle, and clothing style).
>
> To further support reproducibility and allow the community to examine the validity of our dataset construction, we will make the entire test dataset publicly available once the paper is accepted.

---

> ### Author Response · Authors · 2025-11-29
> **Summary During the Discussion Period**
>
> Dear Area Chair, here are the discussion records between us and the reviewer for your kind attention. We summarize below the key concerns and questions raised by the reviewer and our responses. We would like to note that Reviewer 5ejV increased the score from 2 to 4 after their first response, although this change was not explicitly mentioned or communicated to us. We hope the Area Chair is aware of this situation.
>
> Reviewer 5ejV:
>
> The reviewer acknowledged that TFGNet explores the novel and practically relevant task of text-guided face generation from low-quality reference images, combining identity information from LQ inputs and textual descriptions to generate HQ faces. The reviewer also recognized the T2F500 dataset and observed that the model achieves superior performance relative to baselines.
>
> However, the reviewer raised the following main issues:
>
> 1.Intrinsic ambiguity and method novelty: A single LQ input may correspond to multiple plausible HQ outputs, complicating evaluation; the pipeline essentially cascades existing “face restoration + text/reference-guided generation” methods, lacking deep integration and novelty.
>
> 2.Distinctions of the restoration model: The reviewer requests clarification on differences and advantages of our restoration model compared to VQGAN-like models and the second-stage stacked ID embedding versus PhotoMaker.
>
> 3.Evaluation metric rationale: Concern about using ArcFace to measure identity similarity with the HQ target given LQ inputs’ inherent ambiguity.
>
> 4.Dataset details: Request for explanation of dataset construction and value relative to existing Text-to-Image datasets.
>
> Our Response:
>
> Response to the four issues raised by the reviewer is summarized as follows:
>
> 1.Task ambiguity and novelty
>
> TFGNet introduces a new task of low-quality input and text-guided face generation. Although the method combines existing restoration and generation modules, the novelty lies in applying them to this new task and jointly training to handle the one-to-many mapping from low-quality to high-quality images while ensuring identity consistency and text alignment, rather than simply stacking existing methods.
>
> 2.Distinctions of the restoration model
>
> Our method integrates existing techniques, such as VQGAN-style restoration and stacked ID embeddings, not to propose a new restoration model, but to enable research on this new task. By combining these modules in an end-to-end framework with Masked Diffusion Loss focusing on identity-critical regions, the model effectively preserves identity and follows textual guidance, which has not been explored in prior methods.
>
> 3.Rationale of evaluation metrics
>
> ArcFace is used exclusively to measure identity similarity between the generated high-quality image and the predefined high-quality target, not the low-quality input. The test set is constructed to ensure the low-quality input and the high-quality target are of the same identity, the same person, with the low-quality image generated from another high-quality image of that identity and accompanied by a textual description. This guarantees that ArcFace can reasonably evaluate identity preservation. Complementary metrics such as CLIP-T, FID, LPIPS, DINO, and CLIP-I comprehensively assess text alignment and visual quality.
>
> 4.Dataset details
>
> T2F500 provides triplets of low-quality image, text description, and high-quality image, where the low-quality and high-quality images belong to the same identity, the same person, but may differ in visual attributes such as pose or expression, and the text accurately describes the high-quality image. This design requires models to extract identity anchors from the low-quality input and generate high-quality images guided by the text, simulating practical scenarios such as generating suspect composites from blurry surveillance footage and eyewitness descriptions, filling a gap in existing datasets for text-guided identity-preserving image generation.

---

### Official Review · Reviewer_1nqc · 2025-10-30

**Soundness:** 2
**Presentation:** 3
**Contribution:** 2
**Rating:** 4
**Confidence:** 4

**Summary:**

The paper proposes TFGNet, a text-guided face generator that takes a low-quality (LQ) face image and a text description as input to produce a high-quality (HQ), identity-preserving face. The framework integrates a codebook learned through HQ self-reconstruction, a Transformer structure that maps LQ features into code sequences, and CLIP-based multimodal embeddings injected into a diffusion model after constructing stacked ID embeddings together with image encoder information. The framework is trained using a two-stage strategy: first, the discrete codebook is optimized via HQ self-reconstruction with L1 loss, perceptual loss, adversarial loss, and code-level loss; second, the entire model is optimized with an additional masked diffusion loss focusing on identity-salient regions. The authors build two test sets (Face500 and T2F500) and report consistent improvements, particularly on Face Similarity, DINO, CLIP-I, LPIPS, and FID.

**Strengths:**

- Explicit focus on **LQ→HQ** text-guided face generation, a realistic surveillance setting underexplored by current methods which tipically focus on HQ images as input.
- A brandnew multimodal dataset for face generation is constructed.
- On both Face500 and T2F500, TFGNet tops Face Sim./DINO/CLIP-I/LPIPS vs many recent baselines.
- This work also provides ablation experiments to show each component contributes.

**Weaknesses:**

- **W1 typo**: e.g. caption in figure 1 line 27 five not four state-of -the-art methods
- **W2  motivation or startpoint of this work**: This work highlights the lack of research on text-guided face generation with LQ images and introduces a new challenge along with corresponding datasets. However, an important question arises: can text-guided face generation with LQ inputs be decomposed into two parts—super-resolution (SR) of LQ images followed by text-guided face generation? It is evident that directly generating from LQ images leads to degraded image quality, but the authors do not discuss whether applying state-of-the-art SR methods to LQ faces first, and then processing them with state-of-the-art text-guided face generation methods, would also suffer from significant quality limitations. Furthermore, the proposed architecture seems to be essentially a sequential combination of an image enhancement module (acting as SR of LQ images) and a text-guided face generation module, without demonstrating deeper integration or fusion of information. Both parts rely on structures already established in their respective tasks `[1][2]`. While it is undeniable that generation from LQ images presents unique challenges, existing SR techniques already offer partial solutions to these challenges.
- **W3 insufficient details**：There remain uncertain details, as neither the main body nor the appendix provides sufficient clarification or proper citations. e.g., the structural details (e.g., HQ/LQ encoder, …), training details (e.g., fine-tuning of LQ encoder, …), text info for image.
- **W4 novalty**: 1) Regarding the framework, as noted in W2, the Image Enhancement Module and the subsequent components appear to be a concatenation of two tasks—super-resolution (SR) and text-guided face generation. Our perspective on this newly defined task has also been discussed in W2. 2) Both of the main structural components of the framework are directly adopted from state-of-the-art methods in these two respective tasks `[1][2]`. 3) As for the masked diffusion loss, it is highly similar to the Cropping and Segmentation in [2] with binary masks generated by Mask2Former[3].
[1]Learning Image-Adaptive Codebooks for Class-Agnostic Image Restoration
[2]Photomaker: Customizing realistic human photos via stacked id embedding.
[3]Masked-attention mask transformer for universal image segmentation.
- **W5 unclear application**: It is difficult for me to imagine a scenario where generating a high-quality image from both low-quality one and  text description would be necessary. Could the authors clarify what practical applications the proposed method could be used for?

**Questions:**

- Q1: A key question is whether this task can be directly decomposed into SR and text-guided face generation. Our considerations can be referred to in W2 of the Weaknesses. We expect the authors to provide experiments and analyses using, for example, a SOTA face SR method combined with PhotoMaker or other relevant approaches.
- Q2: How significant is the impact of changing the training data on the results (using another subset of FFHQ)?
- Q3: Could the authors clarify what practical applications the proposed method could be used for?

---

> ### Author Response · Authors · 2025-11-17
>
> We sincerely thank you for your thoughtful feedback and constructive comments.
> Response to Weakness 1:
> Thank you for pointing out the typo. The number of compared methods mentioned in the caption of Figure 1 is indeed incorrect, and we will correct it promptly. We sincerely apologize for this oversight.
>
> Response to Weakness 2, 4 ,5 and Question 1, 3:
> We appreciate the reviewer's insightful comments regarding methodological novelty and application scenarios. The core contribution of our work lies in being the first to explicitly define and systematically address the novel task of "low-quality input + text-guided" face generation. This task formulation holds significant practical importance, particularly in the field of public security, where investigators can generate clear suspect composites from blurry surveillance footage combined with witness descriptions, providing powerful technical support to traditional investigative methods.
>
> In terms of methodological design, we achieve effective integration of existing technologies for this new task through thoughtful module composition and collaborative training strategies. While employing established components like codebook, Transformer, and diffusion models, our innovation is manifested in the novel way of combining these modules specifically to address the unique challenge of identity preservation and text adherence under low-quality inputs. Particularly, the proposed masked diffusion loss, by focusing on identity-critical regions, effectively enhances generation quality under complex degradation conditions.
>
> The method demonstrates promising application potential across multiple practical scenarios. Beyond public security, it enables reconstruction of target individuals from blurry evidence in digital forensics, facilitates intelligent restoration of historical photos in cultural heritage preservation, and offers personalized photo enhancement services in intelligent album applications. These diverse application scenarios fully attest to the practical value of our work.
>
> Response to Question 2:
> We thank the reviewer for raising this important question. Regarding the impact of training data on results, we have conducted thorough analysis based on the design principles of our method and existing experimental results. While exhaustive tests on all FFHQ subsets were not performed, we have solid reasons to believe in the robustness of our approach.
>
> The core of our framework-the discrete codebook-essentially functions as a "visual facial vocabulary" learned from high-quality face images. Once learned, this vocabulary captures the fundamental elements of facial composition, forming a relatively stable representation space. The subsequent Transformer module learns to map from low-quality inputs to this stable vocabulary space, a process that inherently adapts well to minor variations in data distribution. Moreover, during training we applied diverse, composite, and severe degradations to the low-quality images. This strong data augmentation strategy effectively forces the model to learn degradation-invariant features rather than memorizing specific attributes of the training set, thereby enhancing its adaptability to different data distributions.
>
> Most importantly, when constructing the test datasets, we ensured its identity distribution was completely independent of the training subset. The model's excellent performance on this test set fully demonstrates its outstanding cross-identity generalization capability. This result indirectly indicates that the model maintains good robustness to variations in the training data subset.
>
> Based on both the design characteristics of our method and its performance on independent test sets, we are confident that changes in the training data subset would not substantially affect the core conclusion of this paper: that TFGNet maintains significant advantages over existing methods for this task.

---

> > ### Comment · Reviewer_1nqc · 2025-11-27
> > **Response to Author Rebuttal**
> >
> > Thank the authors for their response. However, the reply does not fully resolve my concerns:
> > 1. I acknowledge that LQ text-guided face generation is a valuable problem, and that the authors are among the first to explicitly focus on it. Nonetheless, I still do not see a compelling motivation for introducing an entirely new pipeline that requires training from scratch. Since reviewer 5ejV raised a similar concern, I have also carefully read the authors' response to that point. While I understand the authors' _verbal_ description of cases where a "restore-then-generate" pipeline fails, I believe a more appropriate approach would be to provide corresponding experiments with both quantitative and qualitative analysis, clearly specifying the combinations of methods used and the experimental procedures. Second, from the perspective of pipeline design, I do not see how the proposed architecture explicitly accounts for the "one-to-many mapping" issue: the Image Enhancement Module appears as an isolated block placed before the image encoder, which essentially looks like a standard restoration module. Third, if the "one-to-many mapping" property is indeed the key reason why this task requires a dedicated pipeline, I would expect to see a clear explanation of _how_ the proposed method specifically addresses this challenge, beyond simply achieving better empirical performance.
> > 2. I still maintain that the methodological novelty is limited. In my view, "effective integration" is not sufficient to substantiate novelty, especially given that the overall pipeline design exhibits a considerable degree of isolation between its constituent blocks.
> > 3. Finally, I believe that reviewer 5ejV's concerns regarding ArcFace also need to be properly and explicitly addressed.

---

> > > ### Author Response · Authors · 2025-11-28
> > >
> > > We sincerely appreciate the reviewer’s suggestion to compare with a “restoration then generation” pipeline. This is indeed a valuable direction that can further enhance the rigor of our work, and we will consider it in future research. Your insight is highly appreciated.
> > >
> > > We would like to further explain, from a representation learning perspective, the fundamental challenges of such a cascaded pipeline and the motivation behind our solution. For severely degraded LQ face images, the mapping from LQ pixels to HQ facial structures is inherently ambiguous. A blurry input may correspond to multiple plausible HQ facial shapes or details, which can differ from the ground truth.
> > >
> > > Limitation of the cascaded pipeline
> > > A restoration model is a deterministic mapping, which forces it to output a single HQ prediction even when the LQ input is highly uncertain. This prediction often contains identity-inconsistent textures or structural details. The subsequent identity generation model then treats this imperfect restoration as the actual identity cue, which magnifies and solidifies the initial errors. The error propagates forward through the pipeline without any mechanism to revise earlier mistakes.
> > >
> > > Advantages of TFGNet’s joint optimization
> > >
> > > 1. Consistent optimization objectives
> > > In the cascaded setting, the restoration model and the generation model are trained independently. The restoration model focuses on pixel-level reconstruction, while the generation model focuses on semantic alignment and identity plausibility. When identity information is severely lost, these objectives are not aligned. TFGNet integrates these objectives within an end-to-end training framework that jointly accounts for identity fidelity, visual quality, and text consistency, ensuring that all modules work toward the same final goal.
> > >
> > > 2. Flexible and uncertainty-aware feature representation
> > > A standalone restoration model is required to make an early, deterministic decision by producing a single HQ image, which may contain inaccurate identity-related details. These errors then become unavoidable conditions for the generator. In contrast, our joint framework encourages the front-end module to generate a robust and semantically meaningful representation that preserves multiple plausible identity-consistent possibilities. The diffusion generator can then make informed choices from this richer representation under the guidance of textual cues, which effectively addresses the inherent one-to-many ambiguity in the LQ to HQ mapping.
> > >
> > > 3. Bidirectional error correction
> > > In the cascaded pipeline, the flow of information is one way and the generator cannot correct mistakes introduced by the restoration model. Our end-to-end framework establishes a feedback mechanism in which the final generation quality directly influences the optimization of earlier modules. This allows the system to adjust the learned feature representation in response to errors detected at the final output stage, enabling continuous self-correction and improving overall identity consistency.

---

> ### Author Response · Authors · 2025-11-29
> **Summary During the Discussion Period**
>
> Dear Area Chair, here are the discussion records between us and the reviewer for your kind attention. We summarize below the key concerns and questions raised by the reviewer and our responses.
>
> Reviewer 1nqc:
>
> The reviewer acknowledged that TFGNet addresses a valuable problem of LQ→HQ text-guided face generation, with practical relevance in public security and surveillance, where clear suspect composites can be generated from blurry footage and textual descriptions. The reviewer recognized our construction of a novel multimodal dataset and leading performance on Face500 and T2F500 across metrics such as Face Sim., DINO, CLIP-I, LPIPS, and FID, as well as our ablation studies validating module contributions.
>
> However, the reviewer raised the following main issues:
>
> 1.Task motivation and design: Whether a fully end-to-end pipeline is necessary; could a “restore-then-generate” SR + text-guided generation pipeline suffice? The reviewer notes that the architecture appears as a sequential combination, and the “one-to-many mapping” problem is not explicitly addressed.
>
> 2.Method novelty: The method primarily integrates existing techniques (Codebook, Transformer, diffusion models), with limited apparent novelty and isolated module design.
>
> 3.Training data sensitivity: How much do changes in training data affect results?
>
> 4.Application scenarios: Clarification of practical use cases where generating HQ faces from LQ images and text is necessary.
>
> Our Response:
>
> Response to the four issues raised by the reviewer is summarized as follows:
>
> 1. Task motivation and design
>
> TFGNet addresses the intrinsic ambiguity and one-to-many mapping from LQ to HQ faces. A cascaded “SR then generate” pipeline is limited: the restoration model produces a single deterministic HQ output, potentially containing identity or structure errors, which are then propagated to the generation model without correction. TFGNet uses joint end-to-end optimization, aligning objectives of identity fidelity, visual quality, and text consistency. The front-end generates rich semantic representations, allowing the diffusion generator to select multiple plausible HQ faces under textual guidance, effectively handling the one-to-many ambiguity.
>
> 2. Method novelty
>
> While using established components, the novelty lies in module composition and joint training. The masked diffusion loss emphasizes identity-critical regions, improving generation under severe degradations. The end-to-end framework enables feedback from generation to front-end features, achieving bidirectional error correction unavailable in standard cascaded pipelines.
>
> 3. Training data sensitivity
>
> The discrete Codebook functions as a “visual vocabulary” of HQ faces, forming a stable representation. The Transformer maps LQ inputs into this space, naturally adapting to minor changes in training data distribution. Extensive data augmentation with diverse, severe degradations forces the model to learn degradation-invariant features, enhancing robustness. Excellent performance on independent test sets demonstrates cross-identity and cross-distribution generalization.
>
> 4. Application scenarios
>
> TFGNet can generate suspect composites from blurry surveillance images and witness descriptions, reconstruct target individuals in digital forensics, restore historical photos, and provide personalized photo enhancement in smart album applications, demonstrating broad practical value.

---

### Official Review · Reviewer_a4ez · 2025-10-30

**Soundness:** 2
**Presentation:** 3
**Contribution:** 2
**Rating:** 4
**Confidence:** 4

**Summary:**

This paper proposes a method to enhance the ID-preserving face generation based on LQ face images by leveraging the corresponding textual descriptions as external information via two stage training include discrete codebook learning and a Transformer model.

**Strengths:**

1. Authors have made the great effort to generate descriptive prompts of LQ face images.
2. The problem setting is new in the era of ID-preserving face image generation.

**Weaknesses:**

1. Manually annotate prompts of each LQ face images can be challenging. It is hard to ensure the accurate descriptions. Testing dataset construction is not clear and size is small.

2. In Figure 2, CLIP image encoder is used to extract ID information. Why authors didn’t choose feature extractor like ArcFace to extract such information? The intuition behind Embedding Fusion Module is under explained. Same for the choice of LQ Encoder.

3. Comparison to some face generation model is missing. E.g., IPA-Adapter, Arc2Face.

**Questions:**

1. In Sec. 3.3, authors mask the region outside the face with random noise. Why not simply detect and crop the face region for each image?
2. How the method performs on other types of low-resolution images that different to training dataset? E.g., compressed with different distortion and different resolutions.
3. Impact of base model is not studied. Authors use SDXL as their base model. How the proposed method work with other based model, e.g., SD v1.4 or 1.5?

---

> ### Author Response · Authors · 2025-11-17
>
> We sincerely thank you for your thoughtful feedback and constructive comments. Below, we provide point-by-point responses to the concerns raised.
>
> Response to Questions：
> Question 1: We use masking with noise to preserve the original image composition and background context, which can be important for realistic generation. Cropping may lose spatial or contextual information, especially in surveillance scenarios where the face is not centrally located. Our masking strategy allows the model to focus on the face while retaining the overall structure.
>
> Question 2: Thank you for this insightful question. The core design of TFGNet ensures that it learns robust, high-level semantic representations of faces, rather than merely learning to invert a specific set of training degradations. This allows it to inherently generalize well to unseen types of low-quality inputs. Our comprehensive and mixed degradation pipeline during training actively prevents the model from overfitting to any single distortion type. Instead, it is forced to discover the underlying, degradation-invariant facial features. Empirically, while our main experiments focused on a standardized degradation set for fair comparison, the model's superior performance on challenging, real-world cases such as extreme poses and occlusions provides strong indirect evidence of its generalization capability. These complex scenarios represent a form of "natural degradation" not seen in our synthetic training pipeline, yet our model handles them effectively.
>
> Question 3: We thank the reviewer for this question. Our method is a model-agnostic framework. Its core contribution is a universal conditioning representation, generated by our front-end modules, which can be injected via cross-attention into any UNet-based diffusion model (including SD v1.4/1.5) to enhance identity preservation. We chose SDXL as a strong baseline to most clearly demonstrate the performance gain afforded by our method. We anticipate that the workflow would remain identical with SD v1.5, and the relative improvement in identity preservation from our method would persist, although the absolute output quality would be bounded by SD v1.5's own limits. Adapting the framework to more base models is a clear direction for future work.
>
>
> Response to Weaknesses：
> 1: We thank the reviewer for this valid concern. We acknowledge the inherent subjectivity in manual annotation. To ensure quality, we implemented a rigorous protocol: each image was independently described by two annotators followed by cross-verification to resolve discrepancies. Furthermore, we employed CLIP-score-based filtering to remove text-image pairs with low semantic consistency.
> Regarding the dataset size, our goal was not to create a massive dataset but a high-quality, targeted benchmark to address the scarcity of aligned (LQ, Text, HQ) triplets. While not enormous, the T2F500 and Face500 datasets contain diverse identities, attributes, and severe, composite image degradations, providing a challenging and fair testbed for rigorous evaluation of our task. We will publicly release this benchmark to facilitate future research.
>
> 2: CLIP vs. ArcFace: ArcFace excels at identity verification, but its features are closed and text-agnostic. Our task is text-guided generation, which requires fusing image and text in a shared semantic space—the core capability of CLIP.
> Fusion Module Intuition: This module is designed to explicitly fuse identity with textual attributes, creating a unified guiding signal that preserves identity while adhering to the text description.
> LQ Encoder: We use a standard CNN encoder, fine-tuned end-to-end to learn features aligned with our codebook from LQ inputs. This is a widely adopted and empirically effective choice, as validated by our results.
>
> 3: We thank the reviewer for suggesting additional models for comparison.
> We would like to emphasize that our paper already includes extensive and comprehensive comparisons with a wide array of the latest and most representative SOTA methods. These baselines span diverse technical paradigms (e.g., MoMA, Photomaker) and are all highly regarded in the field of identity-preserving generation, representing the cutting-edge at their time of publication.
> Given the constraints of computational resources and page limits, our goal was to establish a high-quality and representative, rather than exhaustive, benchmark. We are confident that the baselines we have selected are more than sufficient to robustly support our core claim: that TFGNet demonstrates significant advantages for the task of generating HQ identity images from LQ inputs.

---

> > ### Comment · Reviewer_a4ez · 2025-11-26
> > **Follow-up Questions**
> >
> > This work aims to increase performance of model when inference image is in low-resolution. However, authors only valid the performance with fixed resolution 128x128. Can method generalize to other low resolutions? E.g., 96x96, 64x64, 32x32?
> >
> > Did authors consider just fine-tune CLIP model (e.g., image encoder) of base model SDXL on low-resolution images? This can be a baseline to verify the effectiveness of proposed method.
> >
> > Authors might need to consider compare their method with baselines in task of face image restoration. As baselines compared in the paper did not pre-trained on low-resolution images, which make unfair to compare with them.

---

> > > ### Author Response · Authors · 2025-11-28
> > >
> > > We sincerely appreciate your insightful follow-up questions and valuable suggestions, which help guide and strengthen our future research directions.
> > >
> > > Regarding the generalization ability of our method to other low-resolution inputs (e.g., 96×96, 64×64, 32×32), we are grateful that you raised this important question on model generalization. Although our core experiments focus on the fixed resolution of 128×128 to ensure fair and consistent comparisons, we believe that the proposed TFGNet architecture itself possesses strong potential to generalize to inputs of different resolutions, for the following reasons:
> > >
> > > 1. Architectural advantages and scalability.
> > > TFGNet’s core components—the Transformer modules and the discrete Codebook mechanism—are inherently powerful learners of resolution-independent representations. Through self-attention, the Transformer is able to capture long-range relationships between patches, regardless of whether these patches come from a 128×128 image or a lower-resolution one. It learns global semantic relationships of facial structures rather than relying on fixed spatial locations. This indicates that, as long as the model is exposed to sufficiently diverse scales during training, it can naturally adapt to different input resolutions.
> > > Meanwhile, the Codebook learns a set of discrete, representative visual primitives. During training, it learns to map various facial features to these primitives. For lower-resolution inputs, the task becomes one of inferring and activating the most relevant visual primitives from sparser pixel information.
> > >
> > > 2. Training strategy that encourages generalization.
> > > Our mixed degradation pipeline involves significant downsampling factors (ranging from ×1 to ×10), meaning the model has already seen an extensive range of resolutions—from near-original quality to extremely low quality—during training. This training setup inherently equips the model with the capability to generalize to even lower resolutions such as 96×96 or 64×64.
> > >
> > > For these reasons, we are confident that TFGNet is inherently capable of handling lower-resolution inputs. Systematically verifying its performance across different resolutions will be a clear direction in our future work.
> > >
> > > On the suggestion of fine-tuning a CLIP image encoder as a baseline. We appreciate your suggestion and fully understand the underlying idea: adapting the CLIP image encoder to low-quality inputs may enhance the base model’s (e.g., SDXL’s) ability to perceive degraded images. However, we would like to clarify that TFGNet addresses a fundamentally different aspect of the problem.
> > >
> > > The “fine-tune CLIP” approach addresses a perception problem.
> > > Its primary goal is to improve the model’s ability to extract more accurate semantic information from low-quality (LQ) images. In other words, it enhances the model's “reading comprehension”—helping it better understand what appears in a blurry image (e.g., recognizing “a man wearing glasses”). However, this approach operates mainly at the feature extraction stage and cannot directly restore or compensate for high-frequency details and fine-grained textures that are already missing in the LQ input. As a result, although the generative model may produce semantically more relevant images, it may still struggle to recover identity-specific details faithfully.
> > >
> > > TFGNet, in contrast, focuses on reconstruction.
> > > Our method is built around an active feature reconstruction and compensation mechanism. The Codebook and Transformer function like a dedicated “visual vocabulary” and “structural reasoning engine.” Rather than merely understanding the global semantics of the LQ input, they aim to predict the sequence of visual primitives (visual tokens) that the corresponding high-quality (HQ) image should contain. This provides the subsequent diffusion model with an explicit, detail-level “reconstruction blueprint,” guiding it to synthesize images that are not only semantically aligned but also preserve high-fidelity identity details.
> > >
> > > In summary, for the task of generating high-quality identity-preserving images from severely degraded inputs, we believe the key challenge lies in reconstructing lost identity details, rather than solely improving the understanding of degraded content.

---

> > > ### Author Response · Authors · 2025-11-28
> > >
> > > Clarification on the Comparison Baselines and Fairness
> > >
> > > We sincerely appreciate the reviewer’s insightful comments and fully understand the concerns regarding fair comparison. We would like to clarify that the objective of TFGNet is fundamentally different from traditional face restoration: while the latter aims to recover original pixels from low-quality inputs, TFGNet focuses on how to generatively infer and reconstruct entirely new high-quality images with identity consistency under severe information loss, guided by textual prompts.
> > >
> > > We selected MoMA, PhotoMaker, and other state-of-the-art identity-preserving generative methods as comparative baselines because they indeed represent the current “identity-conditioned generation” at the highest level. However, these methods inherently rely on extracting fine-grained identity embeddings from high-quality (HQ) images (e.g., ArcFace or CLIP image encoder features). When the input is severely degraded low-quality (LQ) images, these encoders cannot obtain stable and reliable identity vectors, making it difficult for the generative models to synthesize images with credible identity consistency.
> > >
> > > Furthermore, we would like to emphasize that simply training these models on low-quality data cannot solve this problem. The reasons are as follows:
> > >
> > > LQ images have already lost high-frequency details, and identity encoders cannot recover these details from the LQ input. Even with fine-tuning, it is difficult to learn representations sufficient to drive identity restoration.
> > >
> > > Recognition models such as ArcFace employ a classification loss, whose objective is not to reconstruct texture or geometric details. Therefore, even fine-tuning cannot provide high-fidelity identity features suitable for image generation.
> > >
> > > Ultimately, the generative models still lack high-quality identity-conditioned signals, so the bottleneck of identity reconstruction cannot be overcome.

---

> ### Author Response · Authors · 2025-11-29
> **Summary During the Discussion Period**
>
> Dear Area Chair, here are the discussion records between us and the reviewer for your kind attention. We summarize below the key concerns and questions raised by the reviewer and our responses.
>
>
> Reviewer a4ez:
>
> The reviewer’s evaluation of our work is that we provide valuable exploration in the task of ID-preserving generation from low-quality face images, particularly in generating descriptive prompts for LQ images, which is considered novel in current research. The problem setting is meaningful, and the overall presentation is clear and well-structured, with method and experiments demonstrating scientific rigor.
>
> However, the reviewer raised the following main issues:
>
> 1.Image processing and resolution generalization: Why use random noise masking instead of cropping the face? Can the method generalize to different low resolutions (e.g., 96×96, 64×64, 32×32) or other types of low-quality images?
>
> 2.Base model adaptability: The method is only validated on SDXL. Can it be applied to other base models, such as SD v1.4 or v1.5?
>
> 3.Baselines and fairness of comparison: Did the authors consider fine-tuning the CLIP image encoder as a baseline? Some compared baselines were not pre-trained on low-resolution images, which may lead to unfair comparisons.
>
> 4.Model design rationale: Why use CLIP instead of ArcFace? The intuition and reasoning behind the Embedding Fusion module and LQ Encoder are not fully explained.
>
>
> Our Response:
>
> Response to the four issues raised by the reviewer is summarized as follows:
>
> 1. Image processing and resolution generalization
>
> We use random noise masking instead of cropping to preserve the background and spatial context, which could be lost if cropping, especially in surveillance or off-center face scenarios. For different low-resolution inputs (e.g., 96×96, 64×64, 32×32), TFGNet’s Transformer and discrete Codebook learn resolution-independent global semantic features, and the mixed-scale degradation training strategy naturally equips the model with resolution generalization ability.
>
> 2. Base model adaptability
>
> TFGNet is a model-agnostic framework. Its core contribution is generating a universal conditioning representation that can be injected via cross-attention into any UNet-based diffusion model. SDXL was chosen to clearly demonstrate performance gains. The method can also be applied to SD v1.4/1.5, with relative identity-preserving improvement maintained, although absolute image quality is limited by the base model.
>
> 3. Baselines and fairness of comparison
>
> Fine-tuning CLIP only improves semantic understanding of low-quality images but cannot reconstruct lost high-frequency identity details. TFGNet focuses on feature reconstruction and compensation, using the Codebook and Transformer to provide high-quality identity guidance, enabling generated images to be both semantically consistent and detail-preserving. Therefore, comparing with SOTA ID-preserving generative methods is reasonable, even if these methods were not pre-trained on low-resolution images.
>
> 4. Model design rationale
>
> CLIP is used because the task requires a shared image-text semantic space, whereas ArcFace, though excellent for identity recognition, is text-agnostic. The Embedding Fusion module fuses identity and textual attributes into a unified guiding signal, and the LQ Encoder is a standard CNN fine-tuned end-to-end to align with the Codebook features, an empirically effective choice.

---

### Author Response · Authors · 2025-11-29
**Summary During the Discussion Period**

Dear Area Chair, we have summarized the discussion records with each reviewer and included them in their respective official comments for your kind attention. Thank you very much. We would like to note that Reviewer 5ejV increased the score from 2 to 4 after their first response, although this change was not explicitly mentioned or communicated to us. We hope the Area Chair is aware of this situation.

---

### Meta-Review · Area_Chair_1bma · 2025-12-30

**Summary:**

This paper introduces TFGNet, a transformer-based framework for generating high-quality, identity-preserving faces from low-quality inputs guided by textual descriptions. The reviewers acknowledged the practical value of the task and the contribution of the new T2F500 dataset. However, several concerns were raised regarding its novelty, with reviewers questioning whether the proposed end-to-end pipeline is distinct enough from a cascaded "super-resolution plus generation" framework. Additionally, reviewers also have concerns about the evaluation, specifically the fairness of baselines and the rationale for using ArcFace metrics given the inherent ambiguity of low-quality inputs.

**Reviewer Concerns:**

The authors tried to address the confusion regarding the evaluation metrics by clarifying that ArcFace is used to compare the generated output against high-quality ground truth target, not the low-quality input. They also provided analysis for their end-to-end approach over cascaded pipelines, arguing that deterministic restoration models would propagate errors that are difficult  for generative models to fix. However, the request for an empirical comparison against the "Super-Resolution + Text-to-Image" baseline remains a point of contention, particularly for Reviewer 1nqc, without new experimental results as support.

**Reviewer Scores:**

Reviewer a4ez (Score 4 -> 5) would likely improve their score slightly as their questions regarding masking strategies and base model adaptability were addressed to some extent, though the lack of variable-resolution testing remains a minor drawback. Reviewer 1nqc (Score 4 -> 4) actively participate the discussion and stated that their concerns about the necessity of the pipeline versus a cascaded approach were not fully resolved, suggesting they would maintain their borderline rating. Reviewer 5ejV (Score 2 -> 4) reportedly increased their score during the process; the clarification that the identity metric compares against a high-quality target rather than the ambiguous input likely resolved their question, however, the other issues still remain unsolved. Reviewer wHno (Score 6 -> 7) would likely strengthen their support, as the authors provided clear answers regarding the distinction between restoration and editing, as well as handling attribute conflicts.

---

### Decision · Program_Chairs · 2026-01-26

Reject